# Inequalities in cancer mortality between people with and without disability: A nationwide data linkage study of 10 million adults in Australia

Yi Yang[1,2,3*], Nina Afshar[4,5], Zoe Aitken[3], Rebecca J. Bergin[4,6,7], Peter Summers[1,2,3], Roger L. Milne[4,5,8], Sue M. Evans[9,10], Anne Kavanagh[3], George Disney[1,2]

1 Centre for Health Equity, Melbourne School of Population and Global Health, University of Melbourne, Melbourne, Australia, 2 Melbourne Disability Institute, University of Melbourne, Melbourne, Australia, 3 Disability and Health Unit, Centre for Health Policy, Melbourne School of Population and Global Health, University of Melbourne, Melbourne, Australia, 4 Cancer Epidemiology Division, Cancer Council Victoria, Melbourne, Australia, 5 Centre for Epidemiology and Biostatistics, Melbourne School of Population and Global Health, The University of Melbourne, Melbourne, Victoria, Australia, 6 Centre for Quality and Patient Safety Research, Institute for Health Transformation, School of Nursing and Midwifery, Faculty of Health, Deakin University, Geelong, Australia, 7 Department of General Practice and Primary Care, and Centre for Cancer Research, Victorian Comprehensive Cancer Centre, University of Melbourne, Melbourne, Australia, 8 Precision Medicine, School of Clinical Sciences at Monash Health, Monash University, Clayton, Australia, 9 Victorian Cancer Registry, Cancer Council Victoria, Melbourne, Australia, 10 School of Public Health and Preventive Medicine, Monash University, Melbourne, Australia

* yang.y2@unimelb.edu.au

## Abstract

### Background

Cancer is a major yet under-recognised contributor to the mortality gap between people with and without disability. Our study aims to quantify these inequalities to inform cancer control efforts to reduce the gap.

### Methods and findings

We used nationally-linked data (2011–2022) to construct a cohort of over 10 million adults in Australia aged 25–74 years. Disability was measured in 2011 Census as requiring assistance in core daily activities and cancer related deaths identified in national death registrations. We estimated age-standardised and age-specific cancer mortality rates, and absolute and relative mortality inequalities (rate differences and ratios) between people with and without disability. The study included 10,414,951 people. Of the 5,403,503 females, 185,801 (3.4%) reported disability; 183,594 of the 5,011,448 males (3.7%) reported disability. Over 93,940,222 person-years (9.2 years on average), 219,257 cancer-related deaths occurred. After age-standardisation, per 100,000 person-years, there were 314 (95% confidence intervals [CI]: 301, 328) more cancer related deaths in females and 410 (95% CI: 394, 427) more in males with disability (1.96 [95% CI: 1.92, 2.00], and 1.83 [95% CI: 1.80, 1.87] times higher, respectively) than those

**Data availability statement:** The data used in this study are from the Person Level Integrated Data Asset (PLIDA). PLIDA combines information on health, education, government payments, income and taxation, employment and population demographics. PLIDA is governed by existing Australian Bureau of Statistics Privacy and Security protections. The access is limited to approved researchers. Therefore, the data used in our study are not available for sharing. Interested researchers can explore data access options with the Australian Bureau of Statistics (https://www.abs.gov.au/about/data-services/data-integration/access-and-services). Author-generated code used in this study is stored in a secure access environment managed by the Australian Bureau of Statistics. It is available from the authors upon request and subject to Australian Bureau of Statistics clearance.

**Funding:** This work was supported by the University of Melbourne Early Career Researcher Grant (YY, https://www.unimelb.edu.au/) and Melbourne Disability Institute Postdoctoral Fellowship (YY, https://disability.unimelb.edu.au/). The funder did not play any role in the study design, data collection and analysis, decision to publish, or preparation of the manuscript.

**Competing interests:** The authors have declared that no competing interests exist.

**Abbreviations:** ABS, Australian Bureau of Statistics; CI, confidence intervals; ICD-10, International Classification of Diseases version 10; PLIDA, Person Level Integrated Data Asset; STROBE, Strengthening the Reporting of Observational Studies in Epidemiology.

without disability. The largest absolute inequalities were for lung cancer in both females and males (67 [95% CI: 60, 73] and 103 [95% CI: 95, 111] more deaths per 100,000 person-years, respectively), followed by breast cancer in females (54 [95% CI: 49, 60] more deaths), prostate cancer in males (31 [95% CI: 26, 36] more deaths), and colorectal cancer in both sexes (30 more [95% CI: 25, 34] deaths in females and 44 [95% CI: 38, 49] more in males). By 5-year age group, lung cancer was the leading contributor to absolute inequalities in females and males aged 35 years and older. In females, across most age groups, breast cancer was the second largest contributor to absolute inequalities, followed by colorectal cancer. In males, colorectal cancer was the second largest contributor across most age groups, with prostate cancer contributing substantially to absolute inequalities in those aged 55 years and older. A substantial proportion of differences in cancer-related deaths between people with and without disability, across most age groups in both females and males were driven by cancers linked to smoking, obesity, and alcohol consumption. We found similar-sized relative inequalities between individuals with and without disability in mortality due to individual cancers in both sexes. The main limitation of the study was that disability status was measured at a single time point.

## Conclusions

People with disability had higher cancer mortality overall and in relation to specific cancers than people without disability. To close the gap, effort should prioritise interventions that work for people with disability across the cancer control pathway.

---

## Author summary

### Why was this study done?

- Deaths due to cancer are a major yet under-recognised public health concern for people with disability.
- This study aimed to find out whether people with disability in Australia die from cancer more often than people without disability.

### What did the researchers do and find?

- We used disability information collected in the Australian Census in 2011.
- We obtained information on date of death and cause of death up to 2022 from death registrations.
- We found higher death rate in people with disability compared with those without due to a range of cancers.
- Much of this higher cancer-related death rate is linked to cancers that can be prevented or detected early through screening programs, and cancers associated with modifiable risk factors like smoking, obesity, and alcohol use.

PLOS Medicine

### What do these findings mean?

- These findings show that people with disability face inequalities in cancer outcomes that could potentially be avoided.

- Further research is needed to understand the causes of these inequalities.

- Together with past evidence, the findings highlight the importance of making cancer prevention, screening and care more accessible and tailored to the needs of people with disability

- Addressing these inequalities could help reduce avoidable cancer-related deaths among people with disability.

- The main limitation was that disability status was recorded only on the 2011 Census day, and we could not track any changes over time.

## Introduction

Advancing equity in cancer outcomes has become a key theme of national cancer control plans [1]. Multiple countries identify people with disability as a priority population [1], including in Australia's new cancer control plan for 2024–2034 [2].

People with disability experience higher levels of social disadvantage such as poverty, lower educational attainment, social exclusion, and unemployment compared to people without disability [3]. Socioeconomic gradient exists in exposure to cancer risk factors such as smoking and obesity, lower participation in cancer screening, and inequitable treatment and care options, all of which could contribute to higher cancer mortality in people with disability [4,5]. This inequality is compounded by inaccessible healthcare for people with disability [6].

A recent systematic review of cancer inequalities faced by people with disability identified two critical gaps in the literature [7]. The first gap was the scarce evidence that quantified inequalities in cancer mortality between people with and without disability: 10 studies examined overall cancer mortality; only 4 examined type-specific cancer mortality [7]. Given the causes and control strategies differ by cancer types, it is important to examine the inequalities in mortality due to specific cancer types, and how they contribute to the overall cancer mortality gap. Most studies identified subpopulations of people with disability using medical diagnosis, such as diagnosis of intellectual disability or severe mental illnesses. The second gap pertained to the scale of inequalities examined. All studies reported inequalities only on a relative scale, with 1.2–2.5 times higher risks of mortality in people with disability relative to people without disability [7]. Without absolute inequality estimates, the similarly sized relative inequalities give limited information to guide priority setting and resource allocation.

In a recent study, we demonstrated that the magnitude of relative inequality in cancer mortality between people with and without disability was smaller than other causes of death, such as mortality due to neurological conditions [8]. However, when examined on the absolute scale, cancer was the leading contributor, accounting for approximately 20% of all additional deaths in people with disability compared to people without disability [8]. This signifies a public health priority for this population. Reporting inequalities only on the relative scale could potentially lead to cancer control efforts not being appropriately prioritised among people with disability.

This study aims to estimate overall and type-specific cancer mortality inequalities between adults with and without disability in Australia, following the recommended practice to report both relative and absolute inequalities to inform cancer control efforts and research priorities.[9]

## Methods

### Study design and data sources

We used the Australian Bureau of Statistics (ABS) Person Level Integrated Data Asset (PLIDA), which holds a range of administrative and research datasets [10]. Data collections are linked at the unit-record level through deterministic and

probabilistic linkage to a 'population spine' that aims to cover all residents in Australia [10]. The spine is constructed using major administrative datasets on Australia's national healthcare insurance scheme (Medicare), income and taxation, and government benefits and payments (see S1 Methods for details) [10].

Fig 1 shows how we constructed a Census-mortality cohort for this study. Data from the 2011 Australian Census of Population and Housing (the Census) and ABS data on death registrations (2011–2022) were linked via the population spine. The Census collected key socio-demographic information on the whole population of Australia on Census night—9th August 2011. The Census is compulsory, and had a response rate of 96.3% in 2011 [11].

Our cohort included Census respondents aged 25–74 years on Census night who responded to the disability question and were linked to the population spine with unique spine IDs. Deaths among the spine-linked cohort were ascertained through death records linked to the same spine IDs. People older than 74 years were excluded to ensure old age was not the primary reason for both disability and mortality. We excluded respondents who were visitors in Australia.

The study was approved by the University of Melbourne Human Ethics Committee (28947). The requirement for individual participant consent was waived by the University of Melbourne Human Ethics Committee.

## Disability status

The Census disability questions asked about whether the respondent ever needed someone to help with, or be with them, for three core activity areas (self-care, body movement, or communication) and the reasons for needing assistance or supervision. If a person was unable to complete the questions themselves, a household member answered on their behalf. ABS releases a summary variable of disability for research purpose. People were classified as having disability if they reported requiring assistance or supervision with one or more core daily activities due to long-term health conditions or disability that lasted six months or more. People were categorised as non-disabled if they had reported no need for

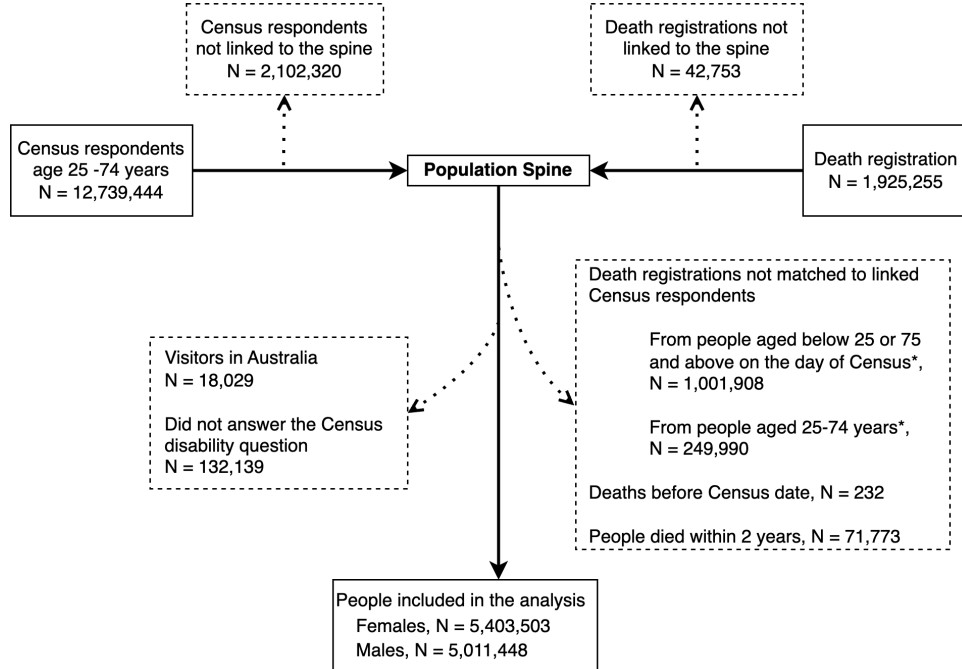

**Fig 1. Flowchart of data linkage and cohort construction of Census 2011 respondents and death registrations from 2011 to 2022.** Footnote: Dashed boxes: exclusions. * For death registrations not matched to Census respondents, age on the day of Census was calculated using date of death and age at death.

assistance or their need for assistance was because of short-term health conditions, difficulty with English, or young age as the only reason (see Supplementary Methods for details) [12]. These questions were designed to largely align with the 'severe or profound limitation' in core activity measure of disability used in the Survey of Disability, Ageing and Carers in Australia [13]. It aligns with the International Classification of Functioning, Disability and Health model of disability through emphasising the impact of functional limitations on an individual's life [14].

## Mortality

Overall cancer mortality and cancer type-specific mortality were determined using the underlying cause of death, defined according to the International Classification of Diseases version 10 (ICD-10) [15].

We examined overall cancer mortality (C00-C97) and mortality due to cancers targeted by Australia's national screening programs for early detection (breast cancer [C50], colorectal cancer [C18–C20, C26.0], cervical cancer [C53]). In addition, we examined mortality due to other specific cancer types that are most common causes of cancer-related deaths in Australia [16], including lung cancer (C33–C34), prostate cancer (C61) and pancreatic cancer (C25). Based on evidence evaluation done by International Agency for Research on Cancer and World Cancer Research Fund [17–19], we also examined mortality due to groups of cancers where sufficient evidence is available for carcinogenicity from smoking, obesity, and alcohol consumption. See S1 Table for the list of cancers and ICD-10 codes.

## Demographic information

Age and sex were reported in the Census. We created 5-year age groups using age on the day of Census.

## Statistical analysis

We followed the cohort for cancer-related deaths to the earlier of (1) death or (2) 31 October 2022 when death registrations were considered complete in the data for this analysis, excluding the first 2 years of follow-up after the Census date to reduce the possibility that individuals who were close to the end of life may be more likely to report disability. We calculated age-standardised mortality rates for any cancer and specific cancer types per 100,000 person-years of follow-up for females and males with and without disability.

The age-standardised mortality rate was calculated as a weighted average of age-specific mortality rates. The weights were the proportions of people in the corresponding 5-year age group of the standard population. Rather than using common standards such as the Australian Population Standard [20], we chose people with disability as the standard population, which allowed us to answer the question 'How does cancer mortality in people with disability compare to people without disability of the same age profile?' In this standardisation process, our mortality estimates for people with disability were largely unaffected, while the mortality estimates for people without disability were adjusted to what they would be if this group had the same age profile to that of the population with disability. This approach allowed us to produce mortality statistics centred around people with disability that more accurately reflect their real-life experience and inequalities [20]. Moreover, our standard population included both females and males with disability to allow comparison of mortality rates by sex.

Age-standardised mortality rate ratios (a measure of relative inequality) and age-standardised rate differences (a measure of absolute inequality) were calculated using people without disability as the reference, with 95% confidence intervals (CI) obtained from 1,000 bootstrap replicates. We presented rate differences, rate ratios and reference rates using a graphical tool for visualising inequalities [21]. We also calculated mortality rates for each five-year age group to ensure that important differences in age-specific mortality between people with and without disability were not masked in the age-standardised rates.

We conducted the following sensitivity analyses: (1) summarising age structure and disability prevalence in each 5-year age group for people linked and not linked to the population spine, to explore potential biases due to excluding people not linked to the population spine, and (2) estimating overall cancer mortality inequalities when categorising people with

PLOS Medicine

missing disability response as people with disability, and as people without disability, to evaluate the impact of excluding people who did not respond to the disability questions.

All analyses were performed in females and males separately. The statistical software used was Stata 17.0.

This study is reported as per the Strengthening the Reporting of Observational Studies in Epidemiology (STROBE) guideline (S1 Checklist). The study did not have a prospective protocol or analysis plan.

## Results

We followed 10,414,951 adults aged 25–74 years over a mean of 9.2 years (93,940,222 person-years). Of the 5,403,503 females, 185,801 (3.4%) reported disability; 183,594 of the 5,011,448 males (3.7%) reported disability. During the follow-up period, 95,661 cancer-related deaths occurred in females, and 123,596 in males.

Fig 2 shows the age distribution by disability status in both sexes. Both females and males with disability had an older age profile compared to those without disability.

### Age-standardised mortality rates and inequalities

Table 1 shows the crude and age-standardised overall cancer mortality rates in females and males by disability status, and the rate differences and rate ratios between groups with and without disability.

The overall cancer mortality rate was higher in people with disability than people without disability in both sexes. Compared to females without disability, females with disability had an additional 314 cancer-related deaths per 100,000 person-years (95% CI [301, 328]), corresponding to a 1.96 times higher cancer mortality rate (95% CI [1.92, 2.00]). Similarly, males with disability had an additional 410 cancer per 100,000 person-years (95% CI [394, 427]), representing a 1.83 times higher mortality rate (95% CI [1.80, 1.87]) than males without disability.

Fig 3 displays the reference age-standardised mortality rates in people without disability, age-standardised rate differences (absolute inequalities) and age-standardised rate ratios (relative inequalities) for individual cancer types. Point

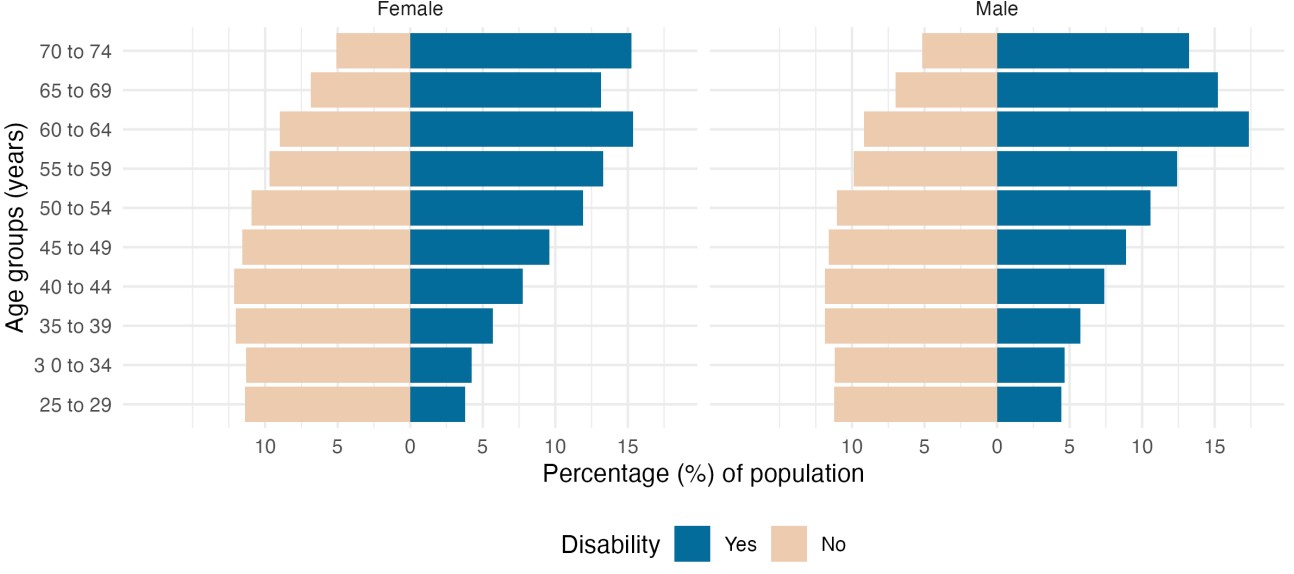

**Fig 2. Age structure of females and males according to disability status, age 25–74 years, 2011, Australia.**

**Table 1. Crude and age-standardised all-cancer mortality rates, rate differences, and rate ratios comparing people with and without disability, age 25–74 years, Australia.**

| Cause of death | People with disability | | | People without disability | | | Rate difference, per 100,000 years (95% CI) | Rate ratio (95% CI)[b] |
|---|---|---|---|---|---|---|---|---|
| | Deaths (person-years) | Mortality rate, per 100,000 person years | | Deaths (person-years) | Mortality rate, per 100,000 person years | | | |
| | | Crude | Age-standardised (95% CI)[a] | | Crude | Age-standardised (95% CI)[a] | | |
| Females | | | | | | | | |
| All cancer | 9,387 (1,534,217) | 612 | 642 (629, 655) | 86,274 (47,445,566) | 182 | 327 (325, 330) | 314 (301, 328) | 1.96 (1.92, 2.00) |
| Males | | | | | | | | |
| All cancer | 12,273 (1,470,272) | 835 | 903 (887, 919) | 111,323 (43,490,168) | 256 | 493 (490, 496) | 410 (394, 427) | 1.83 (1.80, 1.87) |

[a]The Standard Population used is people (both females and males) with disability.

[b]The reference population is people (males and females) without disability.

Abbreviations: CI, confidence interval.

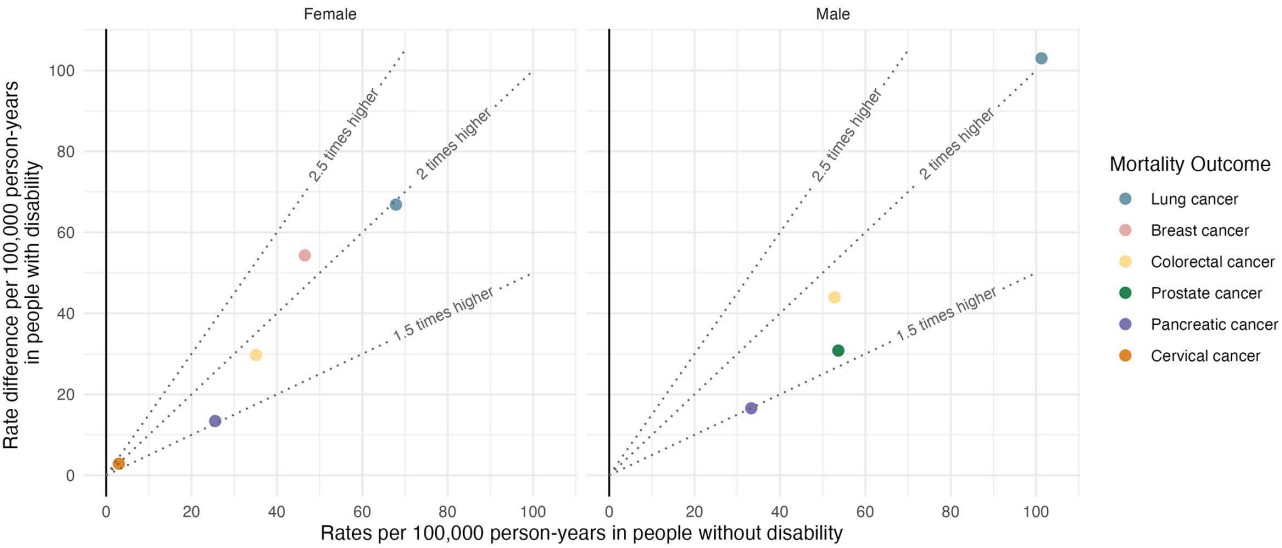

**Fig 3. Age-standardised cancer-specific mortality rate ratio and rate difference comparing females and males with and without disability, age 25–74 years, Australia.** Footnote: Age-standardised mortality rates for people without disability are represented on the x-axis, arranging least to most common underlying causes of death from left to right. The Y-axis displays the rate difference (absolute inequality), indicating additional deaths per 100,000 person-years among people with disability, compared to people without disability with the same age profile. Rate ratios (relative inequalities) are shown by dotted contour lines, each showing how the combination of the reference population mortality rate and the rate difference correspond to a rate ratio. For example, lung cancer-related deaths are common in male without disability, with a reference mortality rate of 101 per 100,000 person-years (x-axis). Lung cancer causes an *additional* 103 deaths per 100,000 person-years in people with disability (y-axis, absolute inequality). This translates to a rate ratio around the '2 times higher' dotted line (relative inequality). This graphical tool allows us to visualise both absolute and relative inequalities in context of the reference rate. Corresponding point estimates and 95% confidence intervals of the age-standardised mortality rate ratios and differences are provided in S2 Table.

estimates and 95% CI were provided in S2 Table. In females, the greatest absolute mortality inequalities were observed for lung cancer (rate difference = 67, 95% CI [60, 73] per 100,000 person-years), breast cancer (rate difference = 54, 95% CI [49, 60]) and colorectal cancer (rate difference = 30, 95% CI [25, 34]), which were also common underlying causes of cancer related deaths in females without disability. Absolute inequalities in mortality due to pancreatic cancer (rate difference = 13, 95% CI [10, 17]) and cervical cancer (rate difference = 3, 95% CI [2, 4]) were moderate. In males, the greatest absolute inequalities were observed for lung cancer (rate difference = 103, 95% CI [95, 111]), which was also the most common underlying cause of cancer-related death in males without disability. This was followed by colorectal cancer (rate difference = 44, 95% CI [38, 49]) and prostate cancer (rate difference = 31, 95% CI [26, 36]). Absolute inequalities for pancreatic cancer were moderate (rate difference = 17, 95% CI [13, 20]). The mortality rate ratios ranged from 1.53 to 2.17 for females and 1.50 to 2.02 for males across individual cancers (S2 Table).

### Age-specific mortality rates and inequalities

S1 and S2 Figs present overall and type-specific cancer mortality rates by five-year age group according to disability status for females and males, respectively (see S3 Table for age-specific rates, rate differences and rate ratios). In both sexes, overall cancer mortality rates increased with age for individuals with and without disability, with consistently higher rates in those with disability. In general, for most cancer types, absolute inequalities widened, and relative inequalities narrowed with increasing age. Absolute inequalities for some cancers were smaller in the oldest age group (70–74 years), including lung and cervical cancers in females, and pancreatic cancer in both sexes.

Fig 4 illustrates individual cancers contributing to the overall cancer mortality rate difference (absolute inequality) by 5-year age group, including screening-related cancers. For both sexes, lung cancer was the leading contributor to the absolute inequalities in mid-aged to older individuals (35 years and older). In females, breast cancer was the second major contributor in nearly all age groups, followed by colorectal cancer. In males, colorectal cancer was the second major contributor across most age groups, with prostate cancer contributing substantially in those aged 55 years and older.

Fig 5 illustrates the proportions of overall cancer mortality rate differences (absolute inequality) driven by cancers related to lifestyle-related exposures, including obesity, alcohol, and smoking. Overall, substantial proportions of the absolute inequalities were driven by deaths due to lifestyle-related cancers, which accounted for 18%–83% (females) and 27%–73% (males) of cancer-related deaths across age groups.

### Sensitivity analysis

The exclusion of people not linked to the spine had negligible impact on the disability prevalences within each the 5-year age group and the age structures of males and females with and without disability (S4 Table). Our sensitivity analysis, which categorised all non-respondents as people without disability, showed virtually no changes in the inequality estimates compared to the main analysis. If the non-respondents were all people with disability, the inequality estimates were slightly attenuated but remained substantial (S5 Table), noting that this is an extreme missing-not-at-random scenario.

### Discussion

We followed over 10 million adults in Australia over 9.2 years on average, and found substantial cancer mortality inequalities between people with and without disability, overall and by cancer type. The largest absolute inequalities in age-standardised rates were observed for lung cancer mortality in both females and males. Substantial absolute inequalities were also found for breast cancer mortality in females, prostate cancer mortality in males, and colorectal cancer mortality in both sexes. For these cancers, which were also common among individuals without disability, we identified sizable relative inequalities, with almost doubled mortality rates for people with disability compared to people without disability of comparable age. Moderate absolute and relative inequalities were seen for mortality from pancreatic cancer, and cervical cancer.

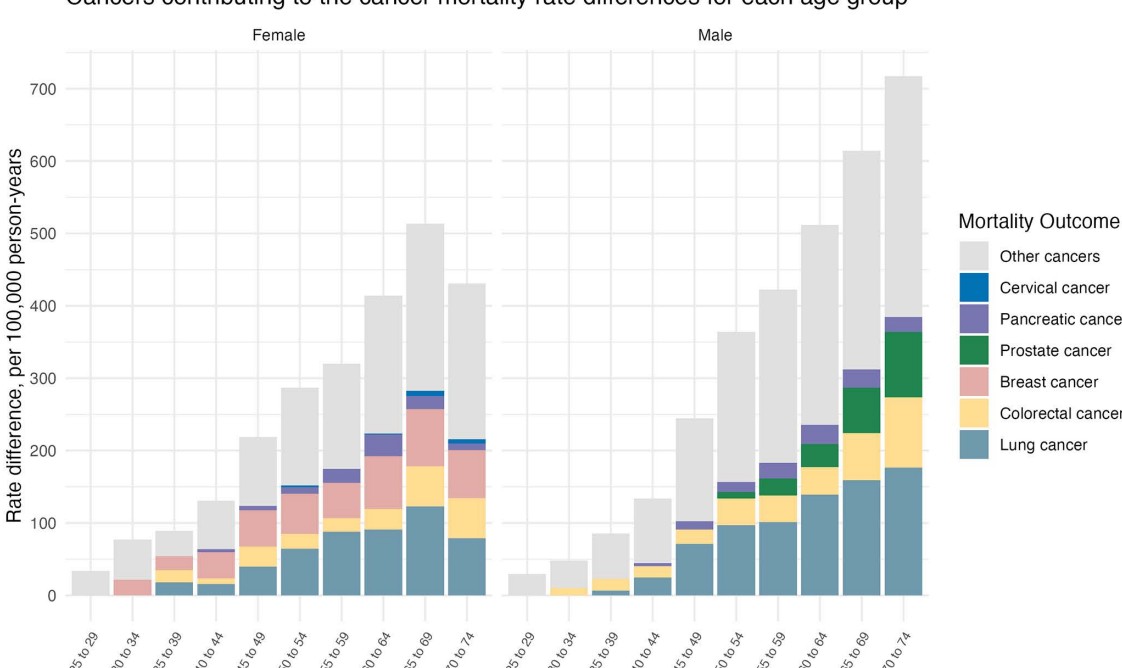

**Fig 4. Specific cancers contributing to the differences in all-cancer mortality rates for people with and without disability in 5-year age groups, age 25–74 years, Australia.** The rate differences underlying this figure are provided in S3 Table. 'Other cancers' include all cancer types not individually presented in the figure.

Relative inequalities in overall cancer mortality narrowed from younger to older age groups as cancer mortality rates increased in the reference populations without disability in both females and males. However, absolute inequalities widened, affecting more people with disability in the older age groups. Breast cancer-related deaths in females over 30 years, prostate cancer-related deaths in males older than 55 years, and lung and colorectal cancers for mid-aged and older individuals in both sexes were substantial contributors to the absolute inequalities in age-specific mortalities. These inequalities were also largely driven by deaths from lifestyle-related cancers, especially those related to smoking.

This study estimated overall and type-specific cancer mortality inequalities, on both relative and absolute scales, using nationally linked electronic data. With close to 95 million person-years of follow-up, we were able to examine, by age group and sex, a range of cancer type-specific deaths, including cancers potentially preventable through screening programs and lifestyle interventions. We followed best practice of reporting inequalities on both relative and absolute scales [9], which has been lacking in much of the previous literature [7]. We used a measure of disability designed to capture its multidimensionality—functional limitation due to health conditions—which differs from medicalised disability definitions based on medical diagnoses used in previous research [14].

Some limitations should be considered when interpreting our results. To quantify inequalities, we inevitably had to use dichotomised disability status—a multifaceted construct. More detailed information on disability was not collected, therefore, we were unable to examine cancer mortality associated with specific disabling conditions, such as leukaemia-related mortality among people with Down syndrome [22]. Disability is also a dynamic status that may change over time. In our study, disability status was measured at a single point in time (the 2011 Census), which captured participants' baseline status but not subsequent transitions. In the 5% sample of 2011 Census records linked to the 2016

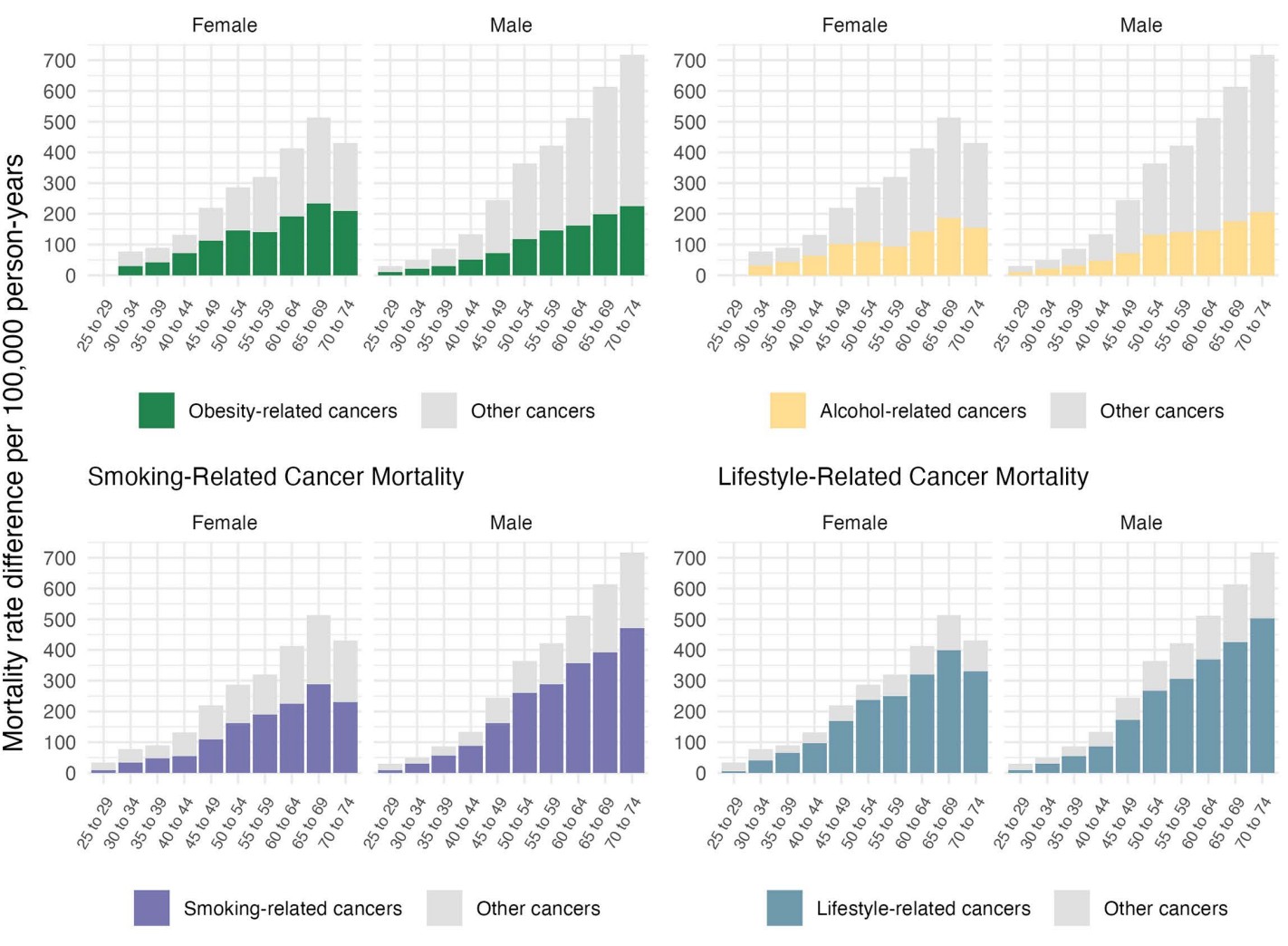

## Cancers contributing to the age-specific overall cancer mortality rate difference

**Fig 5. Lifestyle-related cancers contributing to the differences in all-cancer mortality rates for people with and without disability in 5-year age groups, age 25–74 years, Australia.** The rate differences underlying this figure are provided in S3 Table. Obesity-related cancers, alcohol-related cancers, and smoking-related cancers are classified based on evidence evaluation done by International Agency for Research on Cancer and World Cancer Research Fund. Lifestyle-related cancers are cancers related to obesity, alcohol, or smoking. Cancers related to each lifestyle exposures are not mutually exclusive. See S1 Table for a full list of the cancers and ICD-10 codes. 'Other cancers' include all cancer types not classified as 'obesity-related', 'alcohol-related', 'smoking-related' or 'lifestyle-related' in each of the four panels in the figure.

Census in the Australian Census Longitudinal Dataset, about 33% of those aged 25–74 years who reported a disability in 2011 no longer did so in 2016, while around 3% of those without disability in 2011 reported disability in 2016 [23]. Disability status may also have changed between and beyond the two censuses. Therefore, our results should be interpreted as reflecting inequalities according to baseline disability status, rather than sustained or fluctuating status over time. Future data improvement efforts should aim to reconstruct disability dynamics at a population scale using multiple longitudinal

administrative data sources, such as routine care and disability support data. This would allow better assessment of how time-varying disability influences cancer mortality. Without longitudinal data, we were also unable to examine how certain factors, such as socioeconomic disadvantage, act as causes or consequences of disability. We did not stratify analyses by these factors, as this might introduce bias. For instance, if disability contributed to socioeconomic disadvantage, which in turn led to higher mortality, adjusting for or stratifying by socioeconomic position would underestimate the inequality [24,25]. The Census questions had a strong conceptual basis for identifying individuals with severe functional limitations in performing core daily activities (3.5% of our study population) [13]. Using a broader disability definition that includes less severe disabilities, the estimated prevalence was 21% for ages 25–74 [26]. This may limit the generalisability of our findings to people with less severe forms of disability. However, the disability group in our study is more likely to experience socioeconomic disadvantage, thus higher avoidable inequalities in cancer mortality than people with less severe disability.

Another limitation relates to selecting individuals into our analysis. First, we excluded 2,102,320 (16.5% of all respondents aged 25–74 years) who could not be linked to the population spine, therefore not linked to death data. However, this exclusion is unlikely to have altered our conclusion given its minimal impact on the age structure and age-specific disability prevalence. Second, we excluded people who did not respond to the Census disability questions ($n$ = 132,139, 1% of all respondents aged 25–74 years). However, sensitivity analysis showed that absolute and relative inequalities remained substantial under extreme missing-not-at-random scenarios.

Our findings of relative inequalities aligns with previous studies that found 1.2–2.5 times higher overall cancer mortality rates among subpopulations of people with disability compared to those without, such as people with intellectual disability, psychosocial disability (severe mental illness), and hearing loss [7]. Relative inequalities in mortality were of similar magnitude for specific cancers, such as breast (1.1–1.5 times higher in people with disability) [27–29], colorectal (1.1–1.7 times higher) [27–29], lung and respiratory (1.2 and 1.5 times higher) [27,28], and pancreatic cancer (1.2 and 1.4 times higher) [27,28]. In our study, relative inequalities were larger for these cancers (Fig 3 and S2 Table). This could be partly due to the varying statistical adjustments and focus on more severe forms of disability. Evidence was inconsistent for cervical and prostate cancer [27,29].

Despite previous studies reporting similar relative inequalities to what we found, none reported which cancers caused the most additional deaths in people with disability. Our previous study found that, relative to other causes of death, people with disability had a seemingly moderate 2-fold higher overall cancer mortality compared to people without disability aged 0–74 [8]. This relative inequality translates to 368 (females) and 462 (males) additional cancer-related deaths per 100,000 person-years, making cancer the leading contributor to absolute all-cause mortality inequality among other causes of death [8]. Similarly, despite the comparable magnitude of mortality rate ratios across cancer types in the current study, the absolute inequalities were substantial for lung, colorectal, breast, and prostate cancers, reflecting the high rates in the reference populations without disability.

The mortality inequalities due to cancers for which population screening is available for early detection raises important questions of how we ensure existing screening programs meet the needs of people with disability. In the context of Australia's commitment to eliminating cervical cancer, the 2-fold higher mortality rate among people with disabilities is concerning, even if the absolute inequality is small. Moreover, the adoption of lung cancer screening programs in recent years by multiple countries, including Australia, presents another critical opportunity to reduce the cancer mortality gap between people with and without disability though lung cancer control [30]. Missing this opportunity could potentially widen the cancer mortality gap further. Given the global low uptake of cancer screening programs for people with disability [7,31,32], common barriers need to be addressed to improve participation, including lack of autonomy, inaccessibility, financial cost, and stigma and fear [33,34].

Our findings showed that cancers associated with risk factors such as smoking, obesity, and alcohol consumption contribute substantially to the absolute inequalities in cancer mortality between people with and without disability. These factors are shaped by broader social, environmental, and structural determinants. Studying these modifiable contributors

is important because they represent potential opportunities for interventions to reduce inequalities. Further research, including qualitative research, is also needed to assess whether public health programs are inclusive and effective for people with disability, and to guide equitable strategies for prevention and care.

Smoking-related cancers made a large contribution to the inequalities observed in our study. Research has shown that while smoking prevalence in Australia has declined in the general population, the decline was slower in people with disability, especially those with low incomes [35]. The slower decline suggests that smoking remains as a key driver in the cancer mortality gap between people with and without disability, especially for those experience socioeconomic disadvantage. Socioeconomic disadvantage may manifest in reduced access to timely and appropriate care due to financial barriers and challenges navigating complex health systems [6,36]. For example, a person with disability may delay seeking care due to out-of-pocket costs, inaccessible clinics, or lack of support with communication needs. These barriers could potentially lead to later-stage diagnosis, suboptimal treatment, or lower adherence to follow-up care, all of which contribute to poorer outcomes. Addressing these inequalities requires not only inclusive prevention strategies but also structural changes to make healthcare systems more equitable and accessible for people with disability.

Future studies should report both absolute and relative inequalities, as each provides complementary insights for informing policy, planning, and evaluation. Even modest relative inequalities can translate into a large burden of excess mortality, where targeted actions could deliver significant public health gains. Presenting both measures in future research is also important for monitoring and evaluation of cancer control efforts.

Our study examined inequalities in cancer mortality, which reflect the total disease burden but do not distinguish between differences in cancer incidence and survival. In Australia, cancer screening, diagnosis, and treatment data are not yet linked to disability data. Future data linkage with cancer registries is needed to better understand how disability influences cancer incidence, which may be reduced through prevention, and cancer survival, which can be improved through equitable care.

Our study confirms the need for greater attention to people with disability in cancer research and control. We call for more attention to investigate the mechanisms underpinning the higher cancer mortality rates and identify interventions to close this gap between people with and without disability.

## Supporting information

**S1 Methods. Supplementary Method.**
(DOCX)

**S1 Fig. Age-specific mortality rates due to all cancer and specific cancer types according to disability status, females, age 25–74 years, Australia.**
(DOCX)

**S2 Fig. Age-specific mortality rates due to all cancer and specific cancer types according to status of disability, males, age 25–74 years, Australia.**
(DOCX)

**S1 Table. Lifestyle-related cancers based on assessments of the International Agency for Research on Cancer and the World Cancer Research Fund.**
(DOCX)

**S2 Table. Crude and age-standardised cancer-specific mortality rates, rate differences, and rate ratios comparing people with and without disability, age 25–74 years, Australia.**
(DOCX)

**S3 Table. Age-specific overall cancer and type-specific mortality rates, rate differences and rate ratios comparing people with and without disability, age 25–74 years, Australia.**
(DOCX)

**S4 Table. Age structure and prevalence of disability by 5-year age group in people who were linked and not linked to the population spine.**
(DOCX)

**S5 Table. Age-standardised overall cancer mortality rate differences and rate ratios from sensitivity analysis for missing disability status.**
(DOCX)

**S1 Checklist. The STROBE (STrengthening the Reporting of OBservational studies in Epidemiology) checklist.**
An Explanation and Elaboration article discusses each checklist item and gives methodological background and published examples of transparent reporting. The STROBE checklist is best used in conjunction with this article (freely available on the Websites of PLoS Medicine at http://www.plosmedicine.org/, Annals of Internal Medicine at http://www.annals.org/, and Epidemiology at http://www.epidem.com/). Information on the STROBE Initiative is available at http://www.strobe-statement.org.
(DOC)

## Acknowledgments

We acknowledge the individuals and agencies involved in the collection of Census data used in this study. This study was supported by a University of Melbourne Early Career Researcher Grant and a postdoctoral fellowship by the Melbourne Disability Institute at University of Melbourne.

## Author contributions

**Conceptualisation:** Yi Yang, George Disney.

**Data curation:** Peter Summers.

**Formal analysis:** Yi Yang.

**Funding acquisition:** Yi Yang.

**Investigation:** Yi Yang, Nina Afshar, Zoe Aitken, Rebecca J. Bergin, Peter Summers, Roger L. Milne, Sue M Evans, Anne Kavanagh, George Disney.

**Methodology:** Yi Yang, Nina Afshar, Zoe Aitken, Rebecca J. Bergin, Peter Summers, Roger L. Milne, Anne Kavanagh, George Disney.

**Project administration:** Yi Yang.

**Validation:** Nina Afshar.

**Visualisation:** Yi Yang.

**Writing – original draft:** Yi Yang.

**Writing – review & editing:** Yi Yang, Nina Afshar, Zoe Aitken, Rebecca J. Bergin, Peter Summers, Roger L. Milne, Sue M Evans, Anne Kavanagh, George Disney.

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
