## [Editor Report · Decision Letter 0]

21 Jul 2025

Dear Dr Yang,

Thank you for submitting your manuscript entitled "Inequalities in cancer mortality between people with and without disability: a nationwide data linkage study of 10 million adults" for consideration by PLOS Medicine.

Your manuscript has now been evaluated by the PLOS Medicine editorial staff as well as by an academic editor with relevant expertise and I am writing to let you know that we would like to send your submission out for external peer review.

For clinical studies, please upload a copy of your trial study protocol as a supporting information file. The study protocol should be the version submitted for approval to the institutional review board or ethics committee, should include any amendments to the study protocol, as well as the date of their approval by the institutional review or ethics committee. Please also detail any deviations from the study protocol in the Methods section of your manuscript. The editors will consider the protocol and study conduct prior to a final decision for external review.

Please re-submit your manuscript within two working days, i.e. by Jul 23 2025 11:59PM.

Kind regards,

Jennifer Thorley

Senior Editor

PLOS Medicine

---

## [Decision Letter · Decision Letter 1]

24 Sep 2025

Dear Dr Yang,

Many thanks for submitting your manuscript "Inequalities in cancer mortality between people with and without disability: a nationwide data linkage study of 10 million adults" (PMEDICINE-D-25-02540R1) to PLOS Medicine. The paper has been reviewed by subject experts and a statistician; their comments are included below and can also be accessed here: [LINK]

As you will see, although the reviewers point out that the study addresses an important research question, they also raise critical concerns. After discussing the paper with the editorial team, I'm pleased to invite you to revise the paper in response to the reviewers' comments. We plan to send the revised paper to some or all of the original reviewers, and we cannot provide any guarantees at this stage regarding publication.

We ask that you submit your revision by Oct 15 2025. However, if this deadline is not feasible, please contact me by email, and we can discuss a suitable alternative.

Don't hesitate to contact me directly with any questions (atosun@plos.org).

Best regards,

Alexandra

Alexandra Tosun, PhD

Senior Editor

PLOS Medicine

atosun@plos.org

Comments from the reviewers:

Reviewer #1: This is a strong paper addressing an important and under-explored question - the increased cancer mortality among people with disabilities. It uses a large and well-structured dataset in Australia, which is an almost unique resource. The authors present both absolute and relative differences in cancer mortality, which is an advancement of the topic. There is a relatively stringent definition of disability - requiring assistance or supervision with daily activities - which means that people labelled with disabilities will be severely affected and so the inequities identified may be larger than for all people with disabilities. This limitation is acknowledged.

My main question is the focus on cancers due to lifestyle in the results and discussion. It seems to me that a lot of the inequalities related to screening-related cancers (breast, cervical, prostate, colorectal), which the authors could also consider further.

Moreover, the discussion was relatively long and could be cut by about a quarter. In particular, the section on future research was a little repetitive.

It is a small point - but in the introduction the authors could flag that another reason for higher cancer mortality among people with disabilities is that for specific conditions that are frequently disabling can also lead to cancer - such as Down Syndrome and leukaemia.

Is it possible for the authors to explore further the role of socio-economic differences between people with and without disabilities in cancer mortality inequities? If not, could the authors reference this limitation?

Reviewer #2: "Inequalities in cancer mortality between people with and without disability: a nationwide data linkage study of 10 million adults" analyzes the potential contribution of cancer, to the mortality gap between people with and without disabilities. It was found that people with disability had higher cancer mortality as compared to people without disability, and therefore proposed that interventions targeted towards people with disability should be prioritized across the cancer control pathway.

A strength of the study is its consideration of type-specific effects for cancer, with the reporting of absolute inequality estimates also raised as a novel contribution, that is relevant in resource allocation when compared to other potential contributors (e.g. neurological conditions). Concerns include the simplified definition and self-reported status of disability, and moreover the assignment of participants' disability status based on a single timepoint.

Some further issues might be considered:

1. In the Study Design and data sources subsection, it is stated that data was obtained from the 2011 Australian Census of Population for respondents aged 25 to 74 years (cutoff to minimize age-related disability/mortality) that responded to the disability question, together with ABS data on death registrations from 2011-2022 for mortality.

As supported by the following Disability status subsection, this appears to indicate that disability status is determined at a single timepoint (2011) for all participants. However, it appears likely that disability status may change during the course of the study, especially for older participants. It might therefore be carefully justified as to how this effect is considered, in analyzing and presenting the results. If possible, estimates might be provided for the prevalence of such changes in disability status.

2. Related to the above, it appears likely that disability status is dynamic and correlates with (cancer) mortality. For example, patients determined to be without disability when surveyed in 2011, would appear likely to be eventually considered disabled were they to develop (terminal) cancer afterwards. It might be discussed as to how such effects are addressed in the analysis.

3. In the Study Design and data sources subsection, it is further stated that only persons who responded to the disability question were included in this study. It might be clarified as to the census participation rate (in particular, whether it is mandatory), the disability question response rate (and any implications of the response, e.g. eligibility for disability programs/benefits), and whether this sampling may have been biased (e.g. against persons living in more-remote regions, or disabled persons themselves, etc.)

4. In the Mortality subsection, it is then stated that cancer and cancer type-specific mortality were determined using the underlying cause of death, according to ICD-10. It might be specifically clarified as to whether Australian reporting standards under ABS would include multiple (possible) causes of death, since this is especially material to the analysis.

5. In Figure 2, the presented disability statistics appear to indicate that significantly more than 50% of participants aged above 65 are disabled (as opposed to non-disabled), in 2011; however, SDAC data (https://www.abs.gov.au/statistics/health/disability/disability-ageing-and-carers-australia-summary-findings/latest-release) states that only about 52% of people aged 65 years and over have disability in 2022, despite a general trend towards increasing disability since 2018 (and likely 2011). This seeming large discrepancy might be clarified.

6. In the Statistical analysis subsection, the analysis for age-standardized mortality rates is described. However, it appears that potential covariates (e.g. weight, BMI, socioeconomic status, smoking/drinking/drug use, family history of diseases etc.) relevant to mortality are not accounted for. While the possible impact of socioeconomic disadvantage is discussed as a limitation, the lack of correction for other common covariates/confounders might be addressed. In particular, have previous studies considered these factors?

7. It might be instructive to also present all-cause mortality rates stratified by disability status (as in Supplementary Table 3).

8. The methods used for CI estimation might be briefly described in supplementary material.

9. Minor grammatical errors (e.g. "...compare[d] to those without", Line 291) might be corrected.

Reviewer #3: Thank you for the opportunity to review this interesting paper. This paper provides a clear and concise analysis of cancer mortality among people with disability using Australian mortality data. The analysis is comprehensive and well-documented, covering a wide range of aspects, including interpretations, limitations, and a discussion of inequalities indicators. I recommend that this paper would be published.

Reviewer #4: Thank you for the opportunity to review this national study out of Australia providing a descriptive epidemiology study of cancer mortality among people reporting disability compared to those who did not report disability. There remains an importance of continued documentation of differences in cancer outcomes for adults with disabilities globally.

Major concerns

1. Why use standardization to account for age distribution differences rather than using regression models to directly compare cancer-specific mortality rates over time between the group who reported a disability and the group that did not? As has been pointed out in other studies, using standardization to account for differences in age structure means that you cannot tell whether differences are due to a difference in age specific rates or a difference in population structure. As a result, drawing strong, actionable inferences from the data are not possible.

2. Cancer mortality is a function of both the incidence and fatality of a disease. Cancer mortality estimates or comparisons without accompanying accounts of cancer incidence are less valuable. Adding estimates of cancer incidence alongside the estimates of cancer mortality would significantly strengthen the value and impact of the included work. Otherwise, it is unclear whether differences in mortality rates between those reporting disability and those that did not is caused by a greater burden of the disease or a greater progression from or disparities in management following diagnosis. As a result, drawing strong , actionable inferences from the data are not possible.

3. Disability as defined in this study suggests that disability is not a static construct or medical diagnosis and therefore changes over time, as it uses a timeframe for determining its occurrence (>6 months) and removes folks experiencing more transient forms of disability. However, after that initial differentiation, the definition of disability used in the study assumes no change in disability status over a 15-year period. It's unclear that the measures used in the study have validity or reliability in such an extended time-period and would still be reflected at the time of a cancer-related death. It is also unclear how heterogenous this population is and therefore how to direct future implications or resources, when and to whom to prevent cancer mortality (or risk of developing cancer). As a result, drawing strong, actionable inferences is more challenging.

4. The authors previously published using the same cohort, including a comparison of cancer-related mortality (overall). It is unclear why the authors differently defined the cohort between the two studies, given the rationale provided in the study under review (to ensure disability was not the cause of mortality) and similarity of goals and purposes of the published study and the one under review. Age is a determinant of both cancer incidence and disability, and the authors aim to account for differences in the age distribution between those who report and did not report a disability by standardizing the population to the age distribution of those with a disability. Why would folks additionally need to be excluded on age?

5. The rationale for labeling and focusing on "lifestyle related cancers" inappropriately emphasizes the role of individual factors with the reported disability in cancer-related mortality and places blame or responsibility on the person for their circumstances, rather than the systems or structures in place that may result in that individual's "lifestyle" or cancer outcomes.

---

* Please upload any figures associated with your paper as individual TIF or EPS files with 300dpi resolution at resubmission; please read our figure guidelines for more information on our requirements: http://journals.plos.org/plosmedicine/s/figures. While revising your submission, we strongly recommend that you use PLOS's NAAS tool (https://ngplosjournals.pagemajik.ai/artanalysis) to test your figure files. NAAS can convert your figure files to the TIFF file type and meet basic requirements (such as print size, resolution), or provide you with a report on issues that do not meet our requirements and that NAAS cannot fix.

After uploading your figures to PLOS's NAAS tool - https://ngplosjournals.pagemajik.ai/artanalysis, NAAS will process the files provided and display the results in the "Uploaded Files" section of the page as the processing is complete.

If the uploaded figures meet our requirements (or NAAS is able to fix the files to meet our requirements), the figure will be marked as "fixed" above. If NAAS is unable to fix the files, a red "failed" label will appear above.

When NAAS has confirmed that the figure files meet our requirements, please download the file via the download option, and include these NAAS processed figure files when submitting your revised manuscript.

* Ethics statement: Please provide details on consent.

FIGURES AND TABLES

SUPPLEMENTARY MATERIAL

REFERENCES

STUDY TYPE-SPECIFIC REQUESTS

* Abstract: Please include the study design, population and setting, number of participants, years during which the study took place (enrollment and follow up), length of follow up, and main outcome measures.

* Please ensure that the study is reported according to the GATHER statement (available from https://www.equator-network.org/reporting-guidelines/gather-statement) and include the completed checklist as Supporting Information. Please add the following statement, or similar, to the Methods: "This study is reported as per the Guidelines for Accurate and Transparent Health Estimates Reporting (GATHER) statement (S1 Checklist)." When completing the checklist, please use section and paragraph numbers, rather than page numbers.

* For all observational studies, in the manuscript text, please indicate: (1) the specific hypotheses you intended to test, (2) the analytical methods by which you planned to test them, (3) the analyses you actually performed, and (4) when reported analyses differ from those that were planned, transparent explanations for differences that affect the reliability of the study's results. If a reported analysis was performed based on an interesting but unanticipated pattern in the data, please be clear that the analysis was data driven.

* Please state in the Methods section whether the study had a prospective protocol or analysis plan. If a prospective analysis plan (from your funding proposal, IRB or other ethics committee submission, study protocol, or other planning document written before analyzing the data) was used in designing the study, please include the relevant document(s) with your revised manuscript as a Supporting Information file to be published alongside your study and cite it in the Methods section. A legend for this file should be included at the end of your manuscript. If no such document exists, please make sure that the Methods section transparently describes when analyses were planned, and when/why any data-driven changes to analyses took place. Changes in the analysis, including those made in response to peer review comments, should be identified as such in the Methods section of the paper, with rationale.

---

## [Decision Letter · Decision Letter 2]

9 Dec 2025

Dear Dr. Yang,

Thank you very much for re-submitting your manuscript "Inequalities in cancer mortality between people with and without disability: a nationwide data linkage study of 10 million adults" (PMEDICINE-D-25-02540R2) for review by PLOS Medicine.

Thank you for your detailed response to the reviewers' and editors’ comments. I have discussed the paper with my colleagues, and it has also been seen again by three of the original reviewers. The changes made to the paper were satisfactory to the reviewers. As such, we intend to accept the paper for publication, pending your attention to the editors' comments below in a further revision. When submitting your revised paper, please once again include a detailed point-by-point response to the editorial comments. The remaining issues that need to be addressed are listed at the end of this email.

In revising the manuscript for further consideration here, please ensure you address the specific points made by the editors. In your rebuttal letter you should indicate your response to the reviewers' and editors' comments and the changes you have made in the manuscript. Please submit a clean version of the paper as the main article file. A version with changes marked must also be uploaded as a marked up manuscript file. Please also check the guidelines for revised papers at http://journals.plos.org/plosmedicine/s/revising-your-manuscript for any that apply to your paper.

We look forward to receiving the revised manuscript by Dec 16 2025. However, if this deadline is not feasible, please contact us by email, and we can discuss a suitable alternative.

Sincerely,

Alexandra Tosun, PhD

Senior Editor 

PLOS Medicine

plosmedicine.org

Comments from Reviewers:

Reviewer #2: We thank the authors for addressing our previous comments, particularly the sensitivity analyses on possible census response bias. The descriptive objective of the study is also noted, and might be emphasized in the manuscript.

Reviewer #4: The authors have thoughtfully responded to all questions and concerns. I have no further comments.

Requests from Editors:

GENERAL

* Please confirm that your title complies with to PLOS Medicine's style. Your title must be nondeclarative and not a question. It should begin with main concept if possible. "Effect of" should be used only if causality can be inferred, i.e., for an RCT. Please place the study design ("A randomized controlled trial," "A retrospective study," "A modelling study," etc.) in the subtitle (ie, after a colon).

* Statistical reporting: Please revise throughout the manuscript, including tables and figures.

- Please report statistical information as follows to improve clarity for the reader ""22% (95% CI [13,28]; p</=)"".

- Please separate upper and lower bounds with commas instead of hyphens as the latter can be confused with reporting of negative values.

- Please repeat statistical definitions (HR, CI etc.) for each set of parentheses.

* Please ensure that all abbreviations are defined at first use throughout the text (including statistical abbreviations).

* Please ensure that tables and figures, including those in supplementary files, are appropriately referenced in the main text.

* Please review your text for claims of novelty or primacy (e.g. 'for the first time' or ‘novel’) and remove this language.

* Please confirm that any use of statistical terms (such as trend or significant) are supported by the data, and if not please remove them. The term trend should be used only when the test for trend has been conducted.

* Please define all acronyms used in each figure or table in its corresponding legend.

* Please confirm the use of patient-centered language. Please note that patient-centered language is constructed with the use of post-modified nouns (e.g. 'patients with psoriasis’ (or similar) instead of ‘psoriasis patients’) putting the person first in the sentence structure.

* Please review your manuscript and edit to ensure compliance with our inclusive language requirements https://journals.plos.org/plosmedicine/s/human-subjects-research#loc-categorization

* Please consider including an acknowledgment of the individuals who contributed their data.

* Please state whether you have used any author-generated code in your analysis.

ABSTRACT

* Please confirm that your abstract complies with our requirements, including providing all the information relevant to this study type https://journals.plos.org/plosmedicine/s/submission-guidelines#loc-abstract

* Please quantify the main results (with 95% CIs and p values).

* Please confirm that all numbers presented in the abstract are present and identical to numbers presented in the main manuscript text.

* In the abstract, please include the important dependent variables that are adjusted for in the analyses.

* We suggest changing the language from ‘cancer deaths’ to ‘cancer related deaths’. Please revise throughout.

* We suggest reporting numerators and denominators alongside percentages (as done in the Result section of the main text).

* We suggest reporting the average follow-up time (9.2 years on average) alongside the person-years.

* “We found similar-sized relative inequalities in mortality due to individual cancers in both sexes.” – between individuals with and without disability?

AUTHOR SUMMARY

* In the author summary, in the final bullet point of 'What Do These Findings Mean?', please include the main limitations of the study in non-technical language.

INTRODUCTION

* Please change the heading to ‘INTRODUCTION’.

METHODS AND RESULTS

* We suggested that you complete the GATHER checklist and report your study accordingly, but we cannot find a response to our request in the rebuttal.

* If you do not think that GATHER is the appropriate checklist for your study, please ensure that the study is reported according to the STROBE guideline, and include the completed STROBE checklist as Supporting Information. Please add the following statement, or similar, to the Methods: ""This study is reported as per the Strengthening the Reporting of Observational Studies in Epidemiology (STROBE) guideline (S1 Checklist).""

* Please state in the Methods section whether the study had a prospective protocol or analysis plan. If a prospective analysis plan (from your funding proposal, IRB or other ethics committee submission, study protocol, or other planning document written before analyzing the data) was used in designing the study, please include the relevant document(s) with your revised manuscript as a Supporting Information file to be published alongside your study and cite it in the Methods section. A legend for this file should be included at the end of your manuscript. If no such document exists, please make sure that the Methods section transparently describes when analyses were planned, and when/why any data-driven changes to analyses took place. Changes in the analysis, including those made in response to peer review comments, should be identified as such in the Methods section of the paper, with rationale.

* Figure 2: Please add 'years' as a unit on the y-axis.

* Table 1: For the ‘Rate Ratio’, please add ‘(95% CI)’ to define the statistical meaning of the numbers in brackets.

* “The mortality rate ratios ranged from 1.53 to 2.17 for females and 1.50 to 2.02 for males across individual cancers.” – We suggest referencing S2 Table here (again).

* Figure 4/5: Please convert any stacked bar charts to another data representation for example a table, or other type of graph. At minimum, please provide the values behind the stacked bars in supplementary tables.

* Figure 4/5: Please define ‘Other cancers’.

* Figure 5: In the description, please define ‘Obesity-related cancers’, ‘Alcohol-related cancers’, ‘Smoking-related cancers’ and ‘Lifestyle-related cancers’.

* Please confirm that you provided the unadjusted comparisons as well as the adjusted comparisons in all relevant Tables.

* Please specify the variables controlled for in all relevant Tables.

DISCUSSION

* Please remove the 'conclusions' subheading from the discussion. Please also remove any other subheadings from the discussion.

General Editorial Requests

---

## [Editor Report · Decision Letter 3]

11 Dec 2025

Dear Dr Yang, 

On behalf of my colleagues and the Academic Editor, Wei Zheng, I am pleased to inform you that we have agreed to publish your manuscript "Inequalities in cancer mortality between people with and without disability: a nationwide data linkage study of 10 million adults" (PMEDICINE-D-25-02540R3) in PLOS Medicine.

I appreciate your thorough responses to the reviewers' and editors' comments throughout the editorial process. We look forward to publishing your manuscript, and editorially there are only three remaining points that should be addressed prior to publication. We will carefully check whether the changes have been made. If you have any questions or concerns regarding these final requests, please feel free to contact me at atosun@plos.org.

Please see below the minor points that we request you respond to:

* Ethics statement: Please clarify in the main text whether the requirement for individual participant consent was waived by the University of Melbourne Human Ethics Committee.

* Title: We suggest including the study setting in the title. Editorial suggestion: Inequalities in cancer mortality between people with and without disability: a nationwide data linkage study of 10 million adults in Australia

* Thank you for adding the statement: “Author-generated code used in this study is available from the authors upon request”. We have added the statement to the data availability statement in the online submission form. We strongly encourage you to share your code via a repository that issues persistent identifiers, such as DOIs (e.g. on Zenodo). If you do so, we will be happy to update the Data Availability Statement accordingly.

Before your manuscript can be formally accepted you will need to complete some formatting changes, which you will receive in a follow up email (including the editorial requests above). Please be aware that it may take several days for you to receive this email; during this time no action is required by you. Once you have received these formatting requests, please note that your manuscript will not be scheduled for publication until you have made the required changes.

PRESS

Sincerely, 

Alexandra Tosun, PhD 

Senior Editor 

PLOS Medicine